# Association of Sociodemographic Factors and Maternal Educational Attainment with Child Development among Families Living below the Poverty Line in the State of Ceará, Northeastern Brazil

**DOI:** 10.3390/children10040677

**Published:** 2023-04-03

**Authors:** Hermano A. L. Rocha, Márcia M. T. Machado, Onélia M. M. L. de Santana, Sabrina G. M. O. Rocha, Camila M. de Aquino, Laécia G. A. Gomes, Lucas de S. Albuquerque, Maria D. de A. Soares, Álvaro J. M. Leite, Luciano L. Correia, Christopher R. Sudfeld

**Affiliations:** 1Department of Community Health, Federal University of Ceará, Fortaleza 60430-270, CE, Brazil; 2Laboratory of Epidemiology and Data Analysis, University Health Center ABC, Faculdade de Medicina do ABC, Santo André 09060-870, SP, Brazil; 3Social Protection Secretariat, Ceará State Government, Fortaleza 60130-160, CE, Brazil; 4Integração Serviço, Escola e Comunidade, Unichristus University Center, Fortaleza 60430-275, CE, Brazil; 5Department of Maternal and Child Health, Federal University of Ceará, Fortaleza 60430-270, CE, Brazil; 6Department of Global Health and Population, Harvard T. H. Chan School of Public Health, Boston, MA 02115, USA

**Keywords:** child, educational status, social determinants of health, growth and development

## Abstract

Maternal educational attainment has been identified as relevant to several child health and development outcomes. This study aimed to evaluate the association of sociodemographic and maternal education factors with child development in families living below the poverty line. A cross-sectional study was conducted through telephone contact from May to July 2021 in Ceará, a state in Northeastern Brazil. The study population comprised families with children up to six years of age participating in the cash transfer program “Mais infância”. The families selected to participate in this program must have a monthly per capita income of less than US$16.50. The Ages and Stages Questionnaire version 3 was applied to assess the children’s development status. The mothers reported maternal educational attainment as the highest grade and or degree obtained. The final weighted and adjusted model showed that maternal schooling was associated with the risk of delay in all domains except for the fine motor domain. The risk of delay in at least one domain was 2.5-fold higher in mothers with a lower level of schooling (95% CI: 1.6–3.9). The findings of this study suggest that mothers with higher educational attainment have children with better child development outcomes.

## 1. Introduction

Globally, it is estimated that 250 million children under five years of age in low- and middle-income countries are at risk of failing to reach their full developmental potential due to factors such as poor nutrition, insecurity, and lack of opportunity for early learning [1,2]. Child development has an impact on educational and professional attainment during one’s life course, as it is associated with productivity and income generation in adulthood [3,4,5]. Therefore, child development can contribute to the reduction of intergenerational cycles of poverty [6].

Among low-income families, child development tends to be affected by multiple risk factors [7,8], including inadequate health and nutrition, a home environment with fewer learning opportunities, suboptimal parenting behaviors, poor parental mental health, insecure neighborhoods, and many other factors associated with poverty [9]. On the other hand, maternal education has been reported in the literature as an essential protective factor for child development [10].

Maternal educational attainment has been identified as relevant to several child health and development outcomes (10). Mothers with a higher level of education are more likely to provide greater learning opportunities and may be better able to assess the quality of learning activities [11] as well as make more significant investments in their children’s nutrition and safety [12]. In Brazil, maternal educational attainment has shown significant progress in recent decades, with a gradual decrease in the frequency of women with low educational attainment (defined as less than eight years of school), from 55.4% in 1996 to 18.5% in 2013, and an increase in the frequency of women with high educational attainment, from 7.2% in 1996 to 24.4% in 2013 [11]. However, during the same period, poverty and inequalities persisted. As a result, we aimed to assess the potential protective effect of maternal education on child development in the context of poverty in Brazil.

Despite the advances in income and maternal schooling that Brazil had been experiencing, the COVID-19 pandemic led to economic losses for families, which negatively impacted important factors for child development and nutrition, particularly for vulnerable families with a low socioeconomic status even before the pandemic (15). Nevertheless, maternal education may serve as a buffer to alleviate the potential negative consequences of poverty-related factors on development. Nevertheless, there is little to no data on the relationship of maternal education with child development among families with a low socioeconomic status during the COVID-19 pandemic.

This study aimed to evaluate the association between sociodemographic and maternal education risk factors and child development in families living below the poverty line in Ceará, Brazil, during the COVID-19 pandemic. These results are intended to inform populations at greater risk to help support intervention and program design.

## 2. Materials and Methods

A cross-sectional study was carried out through telephone contact from May to July 2021, during the peak of the second wave of the COVID-19 pandemic in Ceará, a state in Northeastern Brazil. Ceará is a poor state in Northeastern Brazil, with an average per capita income of US$150.00. Subsistence farming is the predominant economic activity in the state’s rural areas, although commerce has become very important in Ceará’s economy, making up more than 70% of the state’s GDP. The state’s estimated population for 2021 was 9,240,580 inhabitants, according to the Brazilian Institute of Geography and Statistics (IBGE), making it the eighth most populous state in the country. We can see Ceará in Figure 1 and the municipalities that took part in the study.

The study population consisted of families with children up to six years of age participating in the cash transfer program “Mais infância”. The families selected to participate in this program should meet three criteria: houses whose walls are made of suboptimal materials (wattle and daub, straw, reclaimed wood) for families in rural areas (the case of rural families), without bathrooms or sanitation, and no running water in at least one room; in addition to a monthly per capita income of less than US$16.50. These families also receive the Brazilian federal government’s conditional cash transfer program (Auxílio Brazil). This population is listed in the program records of the state of Ceará, which had the address and telephone numbers of all families belonging to the target population; a total of 48,000 families were living in the study area. All 48,000 families meet the three inclusion criteria for participation in the “Mais Infancia” program.

For this study, 2000 families were randomly selected for study participation from the list of participants of the cash transfer program “Mais infância” such that all 48,000 families living in the study area are given the same probability of being included in the study. This number was obtained considering events with a prevalence of 5%, a type 1 error of 5%, and a precision of 1%, reaching an estimated 1643. This prevalence is lower than that found in our previous research of 9.4% of developmental delay and 24.7% risk of developmental delay [7]. Interviews were carried out by researchers trained explicitly for this purpose by the research coordination team, using a standardized electronic form to prevent possible input errors. In case of failure to contact a study participant after up to three attempts on three different days, the interviewers called the commercial establishments close to the informed addresses (such as grocery stores) in a last effort to reach the sampled participants. It is important to note that due to the poverty of the target population, an error is expected regarding selected households that will not answer the phone. If, after the attempts described above, the family is not found, it will be considered an absence from the study, which will be accounted for in the statistical analysis (more details in the topic of statistical analysis).

### 2.1. Assessment Tools

To assess the children’s development status, the Ages and Stages Questionnaire version 3 was applied [12], which was validated in Portuguese (ASQ-BR) [13] and has been previously used for research studies in Brazil [14]. This questionnaire consists of a series of questionnaires divided into 20 different age ranges, which seek to evaluate children aged between 2 and 66 months (five and a half years). Five domains of child development are measured in the ASQ-BR subscales: communication, gross motor coordination, fine motor coordination, problem-solving, and personal/social skills [12]. The interviewers were trained for 20 h by medical professionals. This ASQ-BR has been validated for telephone data collection and was shown to have good reliability [15].

The mothers reported maternal educational attainment as the highest grade and/or degree obtained by the mother. This variable was categorized according to the traditional school levels in Brazil, considering graduation in Ensino Fundamental (Junior High School) school as the cut-off point: completion of junior high school (up to 8 years of schooling, 0–8 grades) and high school level (greater than 8 years of schooling, 9–11 grades) (22). This choice was made to ensure comparability with national and international data.

Food insecurity was assessed using the Brazilian Food Insecurity Scale (EBIA, Escala Brasileira de Insegurança Alimentar), which has been validated in Brazil for food security screening and recommended by the Brazilian Ministry of Social Development and Fight against Hunger. In this study, we used the short version of the EBIA, which contains five questions, of which answers vary from never having experienced the measured aspect of food insecurity to experiencing it every day [16]. The Self-Reporting Questionnaire (SRQ-20) was used to assess maternal depression. The SRQ-20 has been validated in several countries, including Brazil [17]. Each SRQ-20 affirmative answer scores a value of 1 to constitute the final score by summing up all answers to all 20 questions. The scores obtained are related to the probability of the presence of a nonpsychotic disorder, ranging from 0 (no probability) to 20 (extreme probability). Additionally, cases with scores greater than eight were considered positive [18]. Information on the families’ socioeconomic status was also collected.

### 2.2. Statistical Analysis

Age- and gender-standardized scores of the ASQ-BR [19] were analyzed for children aged five months or older. For children younger than five months, the US standards were used [20]. Domain scores less <−1 standard deviation were defined as at risk for developmental delay [12,21]. We also considered the presence of delay in any of the X domains as an outcome. Descriptive statistics and prevalence rates of risk of development delay are presented. The association of the main determinants and child development delays in domains was assessed using the χ^2^ test for categorical variables and the Mann-Whitney test for continuous variables.

To estimate the association of sociodemographic factors with each child’s development outcome, generalized estimating equation models were used. Considering the result of the bivariate analysis, we focused the multivariate analysis on the association between child development and maternal education. First, the unadjusted associations were estimated using models in which maternal educational attainment was the only variable. To identify the impact of sample losses on the results obtained, we also performed a sensitivity analysis using inverse probability weighting methods, a type of propensity score analysis. The weights were applied when modeling to estimate the inverse-weighted association between maternal educational attainment and child development outcome. Again, maternal educational attainment was the only variable in the model. Weights are based on the results of an exposure selection model, estimated using logistic regression with maternal educational attainment as the dependent variable and the baseline characteristics—child age, child sex, living in the rural area, maternal age, monthly income, interviewer, food insecurity, and an SRQ score higher than eight—as independent variables. The weights for each participant were calculated as the inverse of the probability of the observed low maternal schooling being conditional on the observed covariates [22]. Finally, the weighted generalized estimating equation model was estimated while additionally controlling for independent variables—child age, child sex, living in the rural area, maternal age, family monthly income, interviewer, food insecurity, and an SRQ score higher than eight—because these factors were found to be associated with child development in our previous research [7,23]. Significance tests and confidence intervals for the estimates from all models were based on robust standard errors. For this analysis, odds ratios and 95%CIs were calculated, and p-values were two-sided. Children with cognitive or motor impairments were excluded from the final analysis in this study. The obtained data were tabulated and analyzed using the software IBM SPSS Statistics for Windows, version 23.0.

### 2.3. Ethics

The study was approved by the Research Ethics Committee of [REMOVED TO DE IDENTIFY]. Free and informed consent was obtained from all research participants, and these were recorded on the online platform and written during the in-person visits.

## 3. Results

### 3.1. Sociodemographic Characteristics

In total, 479 mother-child pairs participated in the study. The mean child age was 40.2 months old (SD: 17.7), and most lived in the rural area (64.4%). In total, 52.4% of the participating mothers had studied only up to eighth grade (high school). On average, they were 31.1 years old, and the highest percentage (34.4%) lived with a partner without being legally married, 81.4% self-declared pardo (grayish-brown), and 66.4% were Catholic. Most mothers also did not have a formal occupation, working as homemakers (81.4%), and 25.4% of the women screened positive for common mental disorders (anxiety and depression), as shown in Table 1.

The socioeconomic status of the families is depicted in Table 1. In more than a third of children (36.4%), the father did not live with the family. The families’ average monthly income was R$527.4 (reais) or approximately US$100 per month. Almost 50% of the houses were constructed of brick masonry with coating, and the majority (44.4%) of the families used water extracted from wells, with 62.4% having access to piped water inside the house and 78.4% having bathrooms. In addition, almost 30% of the families had severe food insecurity.

### 3.2. Child Development Outcome

The standardized scores were around 0 for all assessed domains. When assessing the risk of delay in each of the domains, 11.4% of delay risk in the communication domain, 27.4% in the gross motor domain, 22.4% in the fine motor domain, 16.4% in the problem-solving domain, and 19.4% in the personal/social domain were identified, and when evaluating the presence of delay in any of the domains, a prevalence rate of 50.4% risk of developmental delay was identified. (Table 2).

In the bivariate analysis of the factors modeled as determinants of risk of delay in child development, shown in Table 3, an association with the child’s sex was observed with a higher prevalence of delay in boys in the personal/social domains, and in at least one domain. The lower level of schooling was associated with risk in all domains except for communication domain. Mental disorders in the mothers was associated with a higher risk of delay in the gross motor domain and delay in at least one domain. Finally, income was associated only with at least one domain with delaty (Table 3). 

The results of the models with the sensitivity analysis of inverse weighting for probability are shown in Table 4. The final weighted and adjusted model showed that maternal schooling was associated with the risk of delay in all domains except for the fine motor domain. The risk of delay in at least one domain was 2.5-fold higher in mothers with a lower level of schooling (95% CI: 1.6–3.9). For the communication domain, the chance was 2.35-fold higher (95% CI: 1.2–4.6); for the gross motor domain, 2.68-fold higher (95%CI: 1.6–4.3); for problem-solving, 2.81-fold higher (95% CI: 1.5–5.0); and for the personal/social domain, it was 2-fold higher (95% CI: 1.1–3.4). The other variables in the model showed no association with the assessed outcomes (Table 4).

## 4. Discussion

In this study, we observed that, among families living below the poverty line during the COVID-19 pandemic, maternal schooling of fewer than 8 years was a risk factor for delayed child development in the communication, gross motor, problem-solving, and personal/social domains. Maternal schooling of 8 or fewer years was reported by 52.4% of mothers. 

As expected, considering the sample selection criteria, the study population faced unfavorable socioeconomic situations and a challenging scenario for full child development. The World Bank updated the International Poverty Line in 2015, establishing the amount of US$1.90 as the minimum required for an average adult to gain access to essential means for survival, considering purchasing power parity. This value remained the same at the time the survey was conducted (31). The study data showed that the families received, in total, around US$ 100 on average, which would be less than the poverty line, even if we only consider the mother-child binomial. Additionally, almost 30% of the interviewed families had severe food insecurity. Brazil and the world showed a significant increase in food insecurity with the events associated with the COVID-19 pandemic, and data from national reports estimate that food insecurity increased in general. The latest FAO report on the subject showed that, in Brazil, the number of people with food insecurity was 61.3 million—or almost 30% of the inhabitants of Brazil, which has an estimated population of 213.3 million. Of this total, 15.4 million face severe food insecurity, representing 7.2% (32). The prevalence found in these families was four-fold higher than expected for Brazil. Additionally, according to the mothers’ description, it was identified that 25.4% of the assessed mothers had common mental disorders, a three-fold higher prevalence rate than that found in mothers in general in the state of Ceará, as disclosed by a statewide study carried out in 2017 using the same instrument (33). This increase has also been observed in studies conducted in Ceará and other populations as having been associated with the COVID-19 pandemic (15, 34).

It was observed that nearly half of the mothers had more than eight years of schooling, which is a considerable advancement if one considers that in the 1980s, this figure did not reach 10% (35). This improvement in maternal educational attainment has been observed throughout Brazil (14). Particularly in Ceará, there has been an almost four-fold increase in the percentage of mothers with more than 8 years of schooling in 20 years. Moreover, almost 50% of the families have houses that were constructed of brick masonry with coating, and most have access to piped water at home and bathrooms with toilets. A good part of these improvements in socioeconomic and maternal conditions verified in Brazil accompanied the increase in the conditional cash transfer programs, the most impacting of which was Bolsa Família (36). Despite the increase in maternal educational attainment in Ceara, the studied mothers have not yet accompanied the state statistics, with a 35% higher prevalence of mothers who had only up to 8 years of schooling in the studied sample, or three times higher than that found in the state. In this context, one can observe that the assessed children were exposed to a significant scenario of poverty. The conditions to which the children were exposed possibly justify the findings found in their development domains. The prevalence of risk of delay in at least one child development domain, which was identified in more than half of the assessed children, is more than twice that identified in the population of the state of Ceará in general (24.6%) (7). In spite of the apparently negative effect of social distancing measures to fight the pandemic on child development (37), it is possible that these children already had a greater deficit than that of the general population before the pandemic, due to the poverty conditions in which they live. Several studies have associated poverty with worse cognitive functions, academic success, and social development (9). Possible deficits in access to adequate nutrition, medical care, toys, and other stimulating items, as well as exposure to less adequate parenting, have been pointed out as possible causes of the association between child development and income conditions (38). Additionally, the rate of acute illness in poor children is higher than in non-poor children, which may lead the poor child having more inflammatory and stressful states, all events that lead to child development harm (39). Although these children are exposed to these factors due to generalized poverty, in the present study, the variation in income among these families did not have an impact on the child development domains, as did nutritional deficiency and maternal mental disorders. We hypothesize that the variation of these factors among the subjects is not sufficient to lead to observable effects in this sample. Although these factors were not shown to be associated with child development domains, maternal educational attainment proved to be an important protective factor.

Even with the high prevalence of developmental delay and the observed adverse conditions, maternal educational attainment stood out as an important protective factor against child development delay. These findings are consistent with the robust body of work that has already demonstrated the positive impact of maternal schooling on the re-duction of infant mortality (40, 41), low birth weight (14), and the improvement of overall child health outcomes (42). The mechanisms for these positive effects are varied, such as children’s improved nutritional status, better birth control measures performed by more educated mothers, a higher frequency of equally more educated fathers, greater use of health services by more educated mothers, greater access to information by these mothers, and better financial returns at work for mothers with a higher education (43). The greater effect found for maternal schooling in relation to income is also in agreement with other studies (44). A systematic review that addressed the topic identified that maternal educational attainment has a much greater effect on reducing infant mortality than income, although the two factors are correlated (40). According to Grossman, more educated mothers can both better raise children given the same set of environmental exposures (production efficiency) and are able to allocate available resources in the environment more efficiently (allocation efficiency) [24].

## 5. Conclusions

We believe that the findings of this study enforce our hypothesis that mothers with better education can manage existing resources more efficiently, impacting the outcomes of children’s health and, consequently, their developmental outcomes. To the best of our knowledge, this is one of the first studies to demonstrate the effect of this greater efficiency on child development outcomes in impoverished populations.

This study has several limitations. First, the assessed sample size was lower than the target sample size. However, considering the prevalence of the risk of child development delay found in this population in previous population studies (7), the assessed sample still has a power greater than 80%, and weighting methods based on the inverse of the probability of low maternal schooling were used. Thus, we consider that the results are robust, regardless of this loss. Moreover, as in all cross-sectional studies, one cannot easily establish causal relationships; however, considering that it is possible that most mothers reached the level of schooling before the observed outcome, we believe the temporality criterion can be assumed. Finally, even though we did not use a diagnostic tool for child development delay, the ASQ has been validated in Brazilian populations, and its accuracy was measured and considered to be satisfactory.

The findings of this study suggest that mothers with higher educational attainment have children with better child development outcomes than those children of mothers with a lower level of schooling, even in families living below the poverty line and with an even greater mother‘s limitation of resources due to the pandemic, which we attribute to the greater efficiency in the allocation of resources. This is one of the few studies that demonstrates this phenomenon for the outcome of child development. As a consequence of these findings, we suggest policies aimed at the schooling of poor mothers, improving the level of education, which is still far below that of the general population, and the continuity of policies that have led to the improvement of maternal educational attainment in Brazil in recent years.

## Figures and Tables

**Figure 1 children-10-00677-f001:**
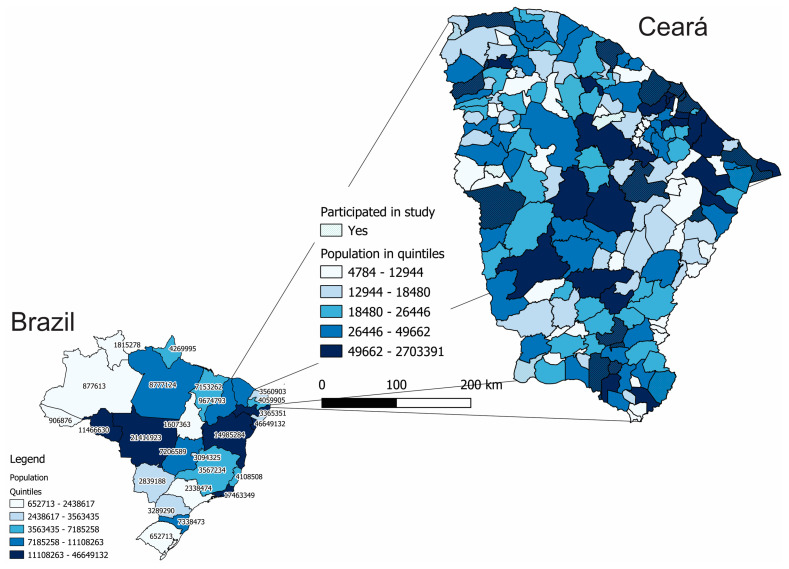
Location of Ceará in northeastern Brazil, with representation of the number of inhabitants by state (in Brazil) and by municipality (in Ceará). Municipalities participating in the *Mais infancia* program are highlighted.

**Table 1 children-10-00677-t001:** Sociodemographic characteristics of the sample of mother-child pairs.

	N (%) or Mean ± Standard Deviation
Child Characteristics	
Child sex	
Female	251 (52.4)
Male	228 (47.4)
Age (months)	40.2 ± 17.7
Residence	
Rural	311 (64.4)
Urban	168 (35.4)
Maternal characteristics	
Maternal age (years)	31.1 ± 7.2
Maternal level of schooling	
More than 8 years	191 (47.4)
Up to 8 years (junior high school)	214 (52.4)
Mother’s marital status	
Lives with a partner in a consensual marriage	142 (34.4)
Single	116 (27.4)
Married	87 (20.4)
Common-law marriage	39 (09.4)
Separated	27 (06.4)
Widowed	4 (0.4)
Race or ethnicity	
Brown	340 (81.4)
White	48 (11.4)
Black	21 (05.4)
Asian descendent	3 (00.4)
Brazilian indigenous	3 (00.4)
Religion	
Catholic	277 (66.4)
Protestant/evangelical	113 (27.4)
No religion	21 (05.4)
Umbanda/Candomblé	2 (00.4)
Spiritist	1 (00.4)
Jehovah’s witness	1 (00.4)
Currently working	
Yes, only at home (homemaker)	339 (81.4)
Yes, outside of home	49 (11.4)
Sim, em casa, para fora	23 (05.4)
I do not work (not even household chores)	4 (00.4)
SRQ score > 8	
No	310 (74.4)
Yes	105 (25.4)
Household characteristics	
Does the father of the child under 6 years old live in the same house as you?	
No	150 (36.4)
Yes	265 (63.4)
In total, how much did the family members earn last month, in reais?	527.4 ± 227.1
What is the predominant material used in the construction of the external walls of your house?	
Brick masonry with coating	207 (49.4)
Brick masonry without coating	133 (32.4)
Wattle and daub with coating	44 (10.4)
Wattle and daub without coating	27 (06.4)
Reclaimed wood	3 (00.4)
Other	1 (00.4)
What type of water supply is used at home?	
Well or spring	183 (44.4)
Piped water	158 (38.4)
Water tank	53 (12.4)
Other	21 (05.4)
Does the household have piped water in at least one room?	
No	155 (37.4)
Yes	260 (62.4)
Is there a bathroom or toilet at home?	
No	91 (21.4)
Yes	324 (78.4)
Did you lose your job during the COVID pandemic?	
No, I did not work	356 (85.4)
Yes, I lost my informal job	52 (12.4)
Yes, I lost my job with a formal contract	7 (01.4)
Has severe food insecurity	
No	291 (70.4)
Yes	124 (29.4)

**Table 2 children-10-00677-t002:** Description of the results of the assessment of child development domains.

	N (%) or Mean ± Standard Deviation
Child development	
Communication score + SD	0.2 ± 0.9
Gross motor score + SD	−0.5 ± 2.2
Fine motor score + SD	0.07 ± 1.6
Problem solving score + SD	0.08 ± 1.1
Personal/social score + SD	−0.06 ± 1.1
Risk of delay in the communication domain	56 (11.4)
Risk of delay in the gross motor domain	131 (27.4)
Risk of delay in the fine motor domain	106 (22.4)
Risk of delay in the problem-solving domain	80 (16.4)
Risk of delay in the personal/social domain	92 (19.4)
Risk of delay in any domain	242 (50.4)

**Table 3 children-10-00677-t003:** Bivariate analysis of the association of the factors proposed as determinants of risk of delay in the child development domains.

Risk of Delay in the Domain:	Communication	Gross Motor	Fine Motor	Problem Solving	Personal/Social	Any Domain
	N (%) or $\overset{-}{x}$ ± DP	N (%) or $\overset{-}{x}$ ± DP	N (%) or $\overset{-}{x}$ ± DP	N (%) or $\overset{-}{x}$ ± DP	N (%) or $\overset{-}{x}$ ± DP	N (%) or $\overset{-}{x}$ ± DP
Lives in the:						
Rural area	35 (11.3)	87 (28.1)	76 (24.5)	53 (17.1)	63 (20.3)	162 (52.3)
Urban area	21 (12.5)	44 (26.2)	30 (17.9)	27 (16.1)	29 (17.3)	80 (47.6)
Child’s sex						
Female	23 (9.2)	57 (22.7)	48 (19.1)	42 (16.7)	38 (15.1)	110 (43.8)
Male	33 (14.5)	74 (32.6)	58 (25.6)	38 (16.7)	54 (23.8)	132 (58.1)
Maternal level of schooling						
More than 8 years	17 (8.3)	38 (19.9)	37 (19.4)	20 (10.5)	30 (15.7)	83 (43.5)
Up to 8 years	32 (15.0)	80 (37.6)	59 (27.7)	52 (24.4)	55 (25.8)	133 (62.4)
Maternal age						
Absence of risk of delay	31.24 (7.24)	31.33 (7.39)	31.39 (7.54)	31.10 (7.57)	31.31 (7.24)	31.31 (7.32)
Risk of delay	30.76 (7.72)	30.81 (7.06)	30.49 (6.39)	31.55 (5.84)	30.65 (7.51)	31.06 (7.27)
SRQ score >8						
No	33 (10.7)	75 (24.3)	67 (21.7)	54 (17.5)	57 (18.4)	150 (48.5)
Yes	17 (16.2)	44 (41.9)	29 (27.6)	19 (18.1)	28 (26.7)	69 (65.7)
In total, how much did the family members earn last month, in reais?						
Absence of risk of delay	527.80 (220.32)	539.89 (232.74)	525.15 (226.51)	532.64 (233.11)	534.25 (228.67)	553.83 (243.07)
Risk of delay	516.98 (270.00)	493.28 (207.65)	530.94 (227.82)	497.79 (191.76)	496.47 (216.85)	502.15 (208.31)
Has severe food insecurity						
No	29 (10.0)	79 (27.1)	71 (24.4)	52 (17.9)	58 (19.9)	149 (51.2)
Yes	21 (17.1)	40 (32.5)	25 (20.3)	21 (17.1)	27 (21.9)	70 (56.9)

Values in bold represent statistically significant values in the bivariate analysis.

**Table 4 children-10-00677-t004:** Multivariate analysis with an inverse weighting of the association of maternal level of schooling up to 8 years with the risk of delay in the child development domains.

	Weighted Unadjusted			Weighted Adjusted *		
	OR	95% CI	*p*-Value	OR ^a^	95% CI	*p*-Value
Risk of delay in any domain	2.26	1.5–3.4	<0.001	2.50	1.6–3.9	<0.001
Risk of delay in the communication domain	1.82	0.9–3.5	0.08	2.35	1.2–4.6	0.01
Risk of delay in the gross motor domain	2.65	1.7–4.3	<0.001	2.68	1.6–4.3	<0.001
Risk of delay in the fine motor domain	1.57	1.0–2.6	0.07	1.63	1.0–2.7	0.06
Risk of delay in the problem solving domain	2.79	1.6–4.9	<0.001	2.81	1.5–5.0	0.001
Risk of delay in the personal/social domain	1.93	1.1–3.2	0.01	2.00	1.1–3.4	0.01

* Adjusted with propensity score analysis. ^a^: adjusted.

## Data Availability

The datasets used and/or analyzed during the current study are available from the corresponding author on reasonable request.

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
