# Peer review of "Association of Sociodemographic Factors and Maternal Educational Attainment with Child Development among Families Living below the Poverty Line in the State of Ceará, Northeastern Brazil"

_children, 2023, doi:10.3390/children10040677_

Round 1

Reviewer 1 Report

Thank you for the opportunity to review the manuscript “Association of sociodemographic factors and maternal educational attainment with child development among families living below the poverty line in the State of Ceará, Northeastern Brazil” for Children. Overall, it was thought provoking and enjoyable read. Generally speaking, I only have positive things to say about this research and don’t have any substantive concerns regarding the manuscript. There is much to like with this paper. My admiration notwithstanding, I have two concerns that should be addressed before moving forward.

·       There is noticeable a lack of contextualization of the case study in the methods section. With consideration to a wider international readership, a map could be helpful to establish the geography of the study region, districts, and spatial interdependencies. Further information could also be given.

·       More information regarding the sampling procedures should be given. Were all 48,000 families living in the study area given the same probability of being included in the study? Did all 48,000 families meet the three inclusion criteria? How much coverage error is there for those who met the three criteria, but did not have a telephone? How was this error accounted for? Without this information, I am currently unable to properly evaluate the results and conclusions.

Author Response

Reviewer 1
Thank you for the opportunity to review the manuscript “Association of sociodemographic factors and maternal educational attainment with child development among families living below the poverty line in the State of Ceará, Northeastern Brazil” for Children. Overall, it was thought provoking and enjoyable read. Generally speaking, I only have positive things to say about this research and don’t have any substantive concerns regarding the manuscript. There is much to like with this paper. My admiration notwithstanding, I have two concerns that should be addressed before moving forward.

  •  

RESPONSE – Thank you very much for your insightful review and for your time dedicated in helping us to improve our manuscript. To the best of our interpretation we fully addressed all comments bellow

There is noticeable a lack of contextualization of the case study in the methods section. With consideration to a wider international readership, a map could be helpful to establish the geography of the study region, districts, and spatial interdependencies. Further information could also be given.

  •  

RESPONSE – We expanded the description of the settings and included a map to best describe Ceará. Please see below and in the marked version of the manuscript.

Ceará is a poor state in Northeastern Brazil, with an average per capita income of US$150.00. Subsistence farming is the predominant economic activity in the state's rural areas, although commerce has become very important in Ceará's economy, making up more than 70% of the state's GDP. The state's estimated population for 2021 was 9,240,580 inhabitants, according to the Brazilian Institute of Geography and Statistics (IBGE) making it the eighth most populous state in the country. We can see Ceará in the figure 1 and the municipalities that took part in the study.

"Please see the figure in the PDF or in the manuscript"

More information regarding the sampling procedures should be given. Were all 48,000 families living in the study area given the same probability of being included in the study? Did all 48,000 families meet the three inclusion criteria? How much coverage error is there for those who met the three criteria, but did not have a telephone? How was this error accounted for? Without this information, I am currently unable to properly evaluate the results and conclusions.

RESPONSE – Thank you for remarking that, we improved the explanation for that questions in the methods section, please see the marked version of the manuscript. Thank you again for your time reviewing our manuscript.

Reviewer 2 Report

In my opinion, the manuscript did focus on the association of sociodemographic factors and maternal education attainment with child development among families in a particular context. And I think the study succeeded to a greater extent in its mission and goal. However, may I suggest to the authors and the editor to look at lines 26-27 in the Abstract. I think the “who” in that sentence refers to the families and not to the children. If I am on the right side, then similar errors should be corrected in other sections so as to improve the quality of the manuscript.

In addition, though the Introduction and References sections look very short, I think the study intended to be specific as possible in its scientific endeavor.

Author Response

Reviewer 2

In my opinion, the manuscript did focus on the association of sociodemographic factors and maternal education attainment with child development among families in a particular context. And I think the study succeeded to a greater extent in its mission and goal.

RESPONSE – Thank you very much for your insightful review and for your time dedicated in helping us to improve our manuscript. To the best of our interpretation we fully addressed all comments bellow.

However, may I suggest to the authors and the editor to look at lines 26-27 in the Abstract. I think the “who” in that sentence refers to the families and not to the children. If I am on the right side, then similar errors should be corrected in other sections so as to improve the quality of the manuscript.

RESPONSE – We adjusted it, sorry. We also double checked all other occurrences.In addition, though the Introduction and References sections look very short, I think the study intended to be specific as possible in its scientific endeavor.

RESPONSE –Thank you again for your time reviewing our manuscript.

Round 2

Reviewer 1 Report

Thank you for the opportunity to review the revised manuscript. The author(s) have addressed all of the concerns raised in my previous review.